METHODS

# Enhancing insights in sexually transmitted infection mapping: Syphilis in Forsyth County, North Carolina, a case study

**Lani Fox**[1,2], **William C. Miller**[3], **Dionne Gesink**[4], **Irene Doherty**[5], **Marc Serre**[1] *

**1** Department of Environmental Sciences and Engineering, Gillings School of Global Public Health, University of North Carolina at Chapel Hill, North Carolina, United States of America, **2** Lani Fox Geostatistical Consulting, Claremont, California, United States of America, **3** Department of Epidemiology, Gillings School of Global Public Health, University of North Carolina at Chapel Hill, North Carolina, United States of America, **4** Epidemiology Division, Dalla Lana School of Public Health, University of Toronto, **5** Julius L. Chambers Biomedical/Biotechnology Research Institute / North Carolina Central University, North Carolina, United States of America

* marc_serre@unc.edu

**Data Availability Statement:** The data are the property of the North Carolina Department of Health and Human Services. They contain sensitive geographic location and demographic data and are

## Abstract

In 2008–2011 Forsyth County, North Carolina (NC) experienced a four-fold increase in syphilis rising to over 35 cases per 100,000 mirroring the 2021 state syphilis rate. Our methodology extends current models with: 1) donut geomasking to enhance resolution while protecting patient privacy; 2) a moving window uniform grid to control the modifiable areal unit problem, edge effect and remove kriging islands; and 3) mitigating the "small number problem" with Uniform Model Bayesian Maximum Entropy (UMBME). Data is 2008–2011 early syphilis cases reported to the NC Department of Health and Human Services for Forsyth County. Results were assessed using latent rate theory cross validation. We show combining a moving window and a UMBME analysis with geomasked data effectively predicted the true or latent syphilis rate 5% to 26% more accurate than the traditional, geopolitical boundary method. It removed kriging islands, reduced background incidence rate to 0, relocated nine outbreak hotspots to more realistic locations, and elucidated hotspot connectivity producing more realistic geographical patterns for targeted insights. Using the Forsyth outbreak as a case study showed how the outbreak emerged from endemic areas spreading through sexual core transmitters and contextualizing the outbreak to current and past outbreaks. As the dynamics of sexually transmitted infections spread have changed to online partnership selection and demographically to include more women, partnership selection continues to remain highly localized. Furthermore, it is important to present methods to increase interpretability and accuracy of visual representations of data.

## Author summary

From 2008 to 2011, Forsyth County, North Carolina saw a dramatic increase in syphilis cases reaching over 35 cases per 100,000, aligning with the state's highly elevated 2021 rate. Our study addresses the challenges of mapping such outbreaks by introducing

federally protected under the United States Government Health Insurance Portability and Accountability Act (HIPAA). They are not routinely available. The NC-DHHS reviews data requests after a formal application for the data. To request access to the data, contact the NC-DHHS Communicable Disease Branch, HIV/STD Prevention and Care Unit by phone at: 919- 733-3419 or mail at: 1905 Mail Service Center, Raleigh, NC 27699-1905; https://www.ncdhhs.gov/divisions/public-health. Access to the data is not guaranteed. The BME numerical processing of these data was performed using MATLAB 7.8 and the BMElib version 2.0c. The code created and used in the analysis can be found at: https://mserre.sph.unc.edu/BMElab_web/mappingStudies/Syp_ForsythNC/. The BMElib code is freely available at: https://mserre.sph.unc.edu/BMElib_web/. For those who do not have programming knowledge there is also a graphical user interface version of BME (BMEGUI) that guides a user through the BME process, download it at: https://mserre.sph.unc.edu/BMElab_web/; code and programs can also be accessed by contacting the University of North Carolina at Chapel Hill Gillings School of Global Public Health, Department of Environmental Sciences and Engineering: https://sph.unc.edu/envr/environmental-sciences-and-engineering-home/.

**Funding:** This study was supported by the National Institute of Allergy and Infectious Diseases, National Institutes of Health (R01 AI067913) to WCM, DG, ID and MS. Additionally, ID also received funding from National Institutes of Health (U54MD012392). All authors received salaries from the above grant to complete the research. The funders had no role in study design, data collection.

**Competing interests:** The authors have declared that no competing interests exist.

innovative methodologies that enhance spatial resolution while preserving patient privacy. Analyzing syphilis surveillance data from the North Carolina Department of Health and Human Services, our work finds the combination of these techniques resulted in more accurate predictions of true syphilis rates, improving accuracy by 5% to 26% over traditional geopolitical mapping methods. The results also identified localized hotspots more effectively, revealing a complex network of transmission emerging from urban endemic areas appearing to spread through sexual core transmitters.

Our study emphasizes the changing landscape of syphilis transmission including shifts in partnership dynamics, showing while online platforms have changed how individuals connect geographic proximity remains a key factor in partner selection. Our study underscores the importance of contextualizing outbreaks within historical and current trends and the continued need for precise visual representations of STI data to aid public health interventions. Overall, this work provides framework for future STI surveillance and response strategies.

## 1. Introduction

Increased understanding of syphilis outbreak development and progression is important as syphilis rates not only persist but are increasing in the United States (US) [1–2] and globally [3–4]. Since reaching a historic low in the US in 2000 and 2001, syphilis has increased nearly yearly and expanded 28.6% from 2020 to 2021 [1–2]. The southeastern region of the US continues to experience higher incidence rates of syphilis than elsewhere in the states [5–13]. Those with ulcerative sexually transmitted Infections (STI), such as syphilis, have an increased likelihood of transmitting or becoming infected with HIV leading to additional concerns about increases HIV morbidity [1–4,14–17]. Additionally, syphilis has continued to be understudied in both the United States and globally [12,18–27], with one article in 2002 titled, "Don't Forget syphilis. . ." [18]. Furthermore, the spread of syphilis is commonly described based on individual-level behaviors, rather than the spatiotemporal or community factors [27]. Thus, even fewer studies track the geographic progression of syphilis outbreaks in the United States [11–12,19–20,27] and additional research on this issue has been advocated for [6,12,19–20,27–28].

As syphilis rates rise in 2022 in North Carolina (NC) very little published research exists on NC outbreaks occurring after 2007. At the end of 2010 Forsyth County, North Carolina (NC) experienced a dramatic, over four-fold increase in syphilis cases and reported the second most cases of early syphilis in the state [14–17]. Similar to previous syphilis outbreaks [5–13,14–17,19–22], escalations in syphilis morbidity during the Forsyth County outbreak of 2008–2011 were present in nearly all demographic groups including those already co-infected with HIV [15–17]. The early syphilis rate in Forsyth County during this outbreak rose to over 35 cases per 100,000 [15–17] similar to the NC state average in 2021 (30 per 100,000) [14].

As shown in Fig 1, in 2022, early syphilis (primary, secondary and early latent syphilis) rates in North Carolina jumped 631% since 2018 [29]. Fig 1 also shows the rise of syphilis in both the US and NC over the past 40 years. Furthermore in 2022, the high-risk demographic groups continue to largely remain the same in North Carolina and the US [1–2,14,29]. Using syphilis surveillance data for Forsyth County, NC from 2008–2011 we analyze this outbreak and use it as a case study to introduce and build upon novel mapping methodologies to better understand STI outbreaks.

When creating STI outbreak maps, protecting patient privacy is essential. In public maps, privacy is commonly protected by aggregating to coarse incidence areas. An incidence area is

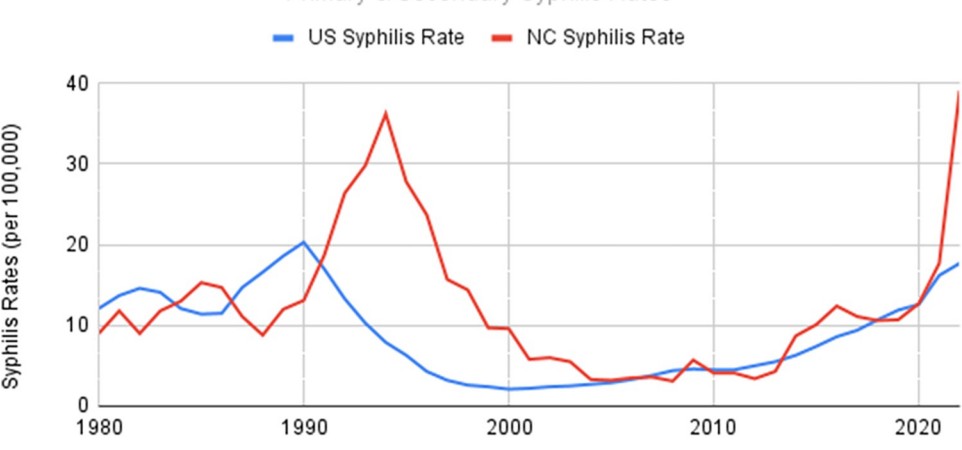

**Fig 1. Annual Primary and Secondary Syphilis Rates from 1980–2022, data compiled from a variety of US Centers for Disease Control (CDC) and NC Department of Human Health Services (NC DHHS) public databases [29–36].**

the geographical foundation on which an incidence rate is defined; the number of new cases in an area over a given period. Most incidence areas are defined using geopolitical boundaries such as states, counties, or census block groups (CBGs). Incidence rates are then assigned to the incidence area's geographical centroid. Although most incidence areas have set geopolitical boundaries, they can also be defined with uniform boundaries with a set shape, such as a circle with centroids based on a regular grid.

Centroid assignment to geopolitical boundaries can obscure pertinent geographical information needed to conduct effective disease mapping [19,37] and cause multiple geostatistical issues leading to inappropriate interpretation of maps. These include the: 1) Modifiable Areal Unit Problem (MAUP); 2) edge effect [19,38–39]; 3) kriging islands of artificially higher and lower incidence at centroids [19]; 4) potentially masked hotspots (hotspots are localized geographical areas with elevated incidence rates) [19,37]; 5) a background rate greater than zero; and 6) the small number problem [19]. MAUP comprises two components: 1) aggregation zone size and 2) areal unit shape [20,38–42]. Even small administrative areas such as census block groups vary greatly in size, shape and underlying population creating a non-uniform distribution of population and geographic centroids. For example within US states, California vs Rhode Island have the same aggregation unit however they have massive differences in both size and population. The edge effect results from displacing an incidence rate to a geographical centroid causing ecological bias. In a map it is shown as hotspots not located in the correct geographic location, the complete loss of a hotspot and/or multiple hotspots incorrectly connected or disconnected to each other [19,38–40].

An increase in spatial resolution (i.e. from state to county to zip code) can reduce MAUP and the edge effect and clarify spatial patterns. With a resolution increase, aggregated rates are displaced less from their original, non-aggregated point sources [19–20,40–47]. Geomasking techniques such as donut geomasking used in this study can ensure patient privacy while increasing the spatial resolution necessary for cluster and outbreak detection [28, 40–47]. However with rare diseases such as syphilis, reducing the aggregation scale also introduces errors due to the scarcity of the data.

Scarcity in disease data can create maps dominated by locations with low populations. This produces challenges to interpret incidence rates and their maps because rates in low population areas can appear alarmingly elevated even when a single case is present, known as the "small number problem" [28,48–50]. Concurrently the size of many geopolitical areas such as census tracts or block groups are population dependent, therefore areas with large populations also have small areas and vice versa. This is generally seen as maps with a hotspot over a large area. Multiple smoothing algorithms have been developed to reduce the small number problem improving the interpretable accuracy of incidence rates estimates also referred to as the true or latent rate [19,48,28,51–54]. Uniform Model Bayesian Maximum Entropy (UMBME), an advanced kriging based smoothing method can produce incidence rate estimates with greater accuracy through mathematical penalizing of areas with small populations [19,28,44].

Past STI studies [19] found the edge effect is especially prevalent in the prediction method kriging where rates are highest at the prediction point. This phenomenon is known as "kriging islands". Kriging islands can make maps more difficult to interpret by obscuring the ability to fully understand the progression of an outbreak and how it moves between susceptible communities [19]. Using the Forsyth County 2008–2011 syphilis surveillance data, we advance STI outbreak analysis methods by increasing spatial resolution with donut geomasking and introducing a global moving window approach to minimize MAUP, edge effect and kriging islands. Finally, employing the Uniform Model Bayesian Maximum Entropy (UMBME) [28,38] method moderates the small number problem. We hypothesized combining these methods would create more easily interpretable maps better delineating the geographical extent and connection of clusters [19]. Using the 2008–2011 Forsyth outbreak as a case study for these methods, our study continues to fill in the gap in our geographic understanding of syphilis and other STIs contextualizing it to other outbreaks.

## 2. Methods

### Study population & data preparation

The study used incidence data from the North Carolina Department of Health and Human Services (DHHS) and US Census data. North Carolina health care providers and laboratories are required to report each newly diagnosed case of syphilis to the county health department including diagnosis date, date of disease or symptoms onset, current residence, syphilis disease stage and limited demographic information. Data preparation, geocoding and geomasking were performed on site at the NC-DHHS. Our first step removed all patient demographic information. This study was reviewed by the University of North Carolina Committee on the Protection of the Rights of Human Subjects and was determined to be exempt, based on exemption criterion 4 (IRB 05–3080).

This work focuses on the spatiotemporal analysis of incidence rates in Forsyth County, NC from 2008–2011. We used syphilis diagnosis stage to estimate when a patient acquired the disease. To support the work of the DHHS who both provided the data for the project and supported it, the early syphilis categories (primary, secondary and early latent) were also used in our analysis. Early syphilis is the stages when the syphilis pathogen has highest infectivity for sexual partner transmission. Late latent syphilis is not considered infectious and excluded from the analysis [20,44]. We estimated the date of infection based on the diagnosis date and median latency period for each syphilis disease stage. Primary syphilis, secondary syphilis and early latent syphilis were back-estimated by 45, 90, and 183 days, respectively [60].

Geographically, Forsyth County is one of the smallest of the 100 counties in North Carolina spanning approximately 1,070 km. In 2010 it was also the 5th most population dense with approximately 351,000 residents. Forsyth has 2 major cities, Winston-Salem and High Point, the fourth

and seventh most populous cities in NC. The remainder of the county is primarily rural [55]. Interstate 40 (I-40) also runs through the county. I-40 spans the entirety of the state and is one of the most critical US east-west highways connecting North Carolina to California [56].

Self-reported residential addresses were corrected with Satori Bulk Mailer software [57] to optimize the geocoding match rate of addresses. Case residences were geocoded using ESRI's ArcGIS 9.3.1 [58] and matched to three geographical locators used by the NC-DHHS for geocoding. The first geocoding round used our highest quality and primary locator created by the North Carolina Department of Transportation containing street-level geographical data. Cases not matched using our first locator were then matched using our secondary locator created by the North Carolina Emergency Response System which contains GPS point locations for households. Cases not matched to the first or second locator were then matched to our tertiary locator created using ESRI's 2006 Street Map shapefile [59], which is primarily used for locating residences with outdated street names, prisons, and military bases. Finally, cases with a post office box address were spatially assigned to that post office address. Matching in this method allows for the highest quality geographic data to be linked to the cases while maximizing the number of cases matched to a geolocator.

Approximately 83% of records were successfully geocoded to a location (n = 497 of 601). Cases that were not geocoded (n = 104 of 601) were excluded from the analysis. Of the cases not geocoded, 36 were in 2009 (6%) and 16 in 2010 (3%). Failure to geocode was usually because: 1) no address is reported; 2) non-existent/false addresses reported; 3) street segments are missing locators- most often occurs on rural routes with unavailable E911 addresses; or 4) incorrect/incomplete addresses (misspellings, abbreviated street names and improper use of rural routes). Failure to match may also occur with persons unhoused, out of state, or with military addresses. Data that could not be geocoded appeared to be missing at random without any noticeable patterns thus we believe it only added noise rather than an inherent bias. After geocoding, data were geomasked using the donut method [28,44–45]. Donut geomasking randomly relocates each geocoded data point within a user defined minimum and maximum radii [28].

## Syphilis incidence rates

**Incidence areas.** Two datasets of syphilis incidence rates were created and compared, US census block groups (CBG) and a uniform grid with a moving window (Fig 2). Our first dataset, composed of the traditional, geopolitical boundaries was created in ESRI's ArcGIS 10 [58] using the US 2000 Census populations and census block group boundaries packaged in a shapefile [58]. This file was also used to calculate the CBG centroids. Fig 2 clearly shows the irregular locations of the CBG centroids, size and shape demonstrating a strong potential for MAUP and the edge effect. The uniform grid approach changes the area of aggregation from a geopolitical boundary to a standardized area with a consistent centroid location allowing aggregation without the variability caused by size and shape [38–40]. Furthermore, overlapping the boundaries of the uniform grid area creates a continuous surface, reducing the edge effect [39–41]. We hypothesize kriging islands, MAUP and edge effect created when aggregating to geopolitical incidence groups can be corrected by creating a uniform grid with a moving window of overlapping perfect circles (shown as turquoise dots with blue circles in Fig 2).

To calculate incidence rates, CBG cases were aggregated spatially by census block group boundaries, and temporally with a rolling period of 6 months duration to lessen the small number problem. Population growth was also incorporated into the crude rate calculation through a linear interpolation of the census block group population for all 12-64-year-olds in 2000 and 2007 assuming positive population growth over the time period.

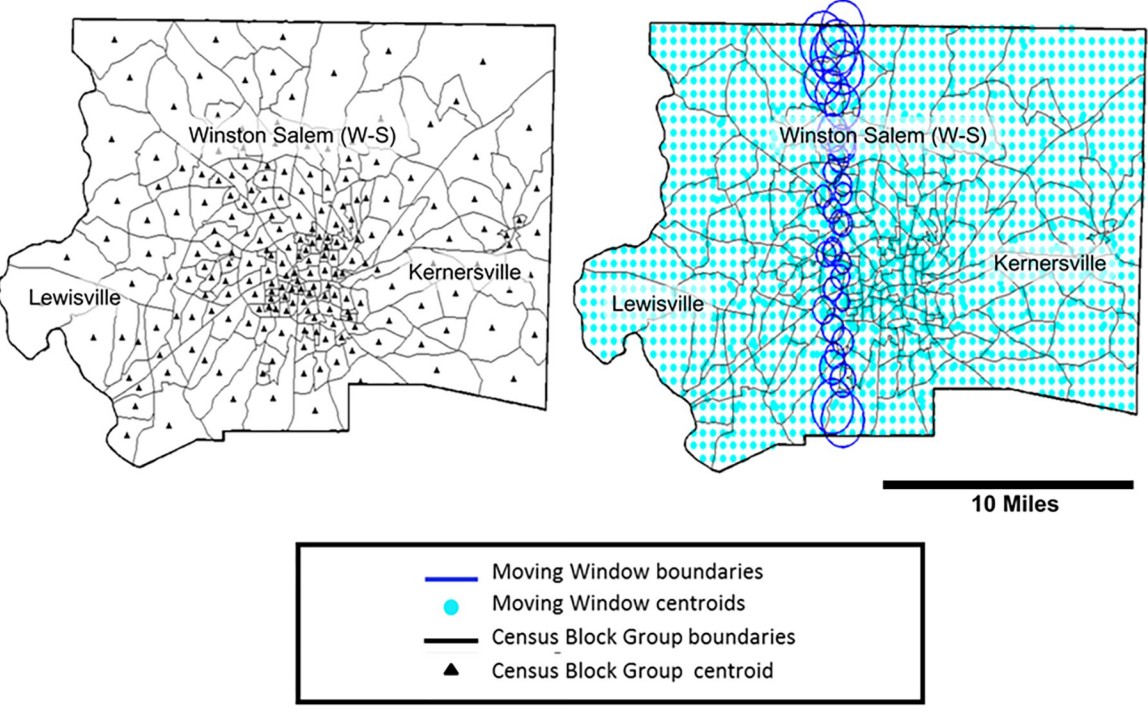

**Fig 2. Geopolitical Census Block group aggregation vs standard grid with a moving window aggregation for incidence areas in Forsyth County, NC.** The CBG boundaries show the non-uniform distribution of geographical centroids whereas the uniform grid/ moving window show an even distribution of centroids and incidence areas size to reducing the edge effect and MAUP. Census Block Group Boundaries are publically available by the US Census at: census.gov/geographies/mapping-files/time-seriesgeo/carto-boundaryfile.2000.html#list-tab-U2GTBOCTYRAP13D9AH.

The crude incidence is denoted as $R_{ij}$ and calculated as $R_{ij} = y_{ij}/(n_{ij}T)$. Incidence period of duration $T$ expressed in years (i.e. $T = 0.5$yr for a 6 month incidence), centered at time $t_j$, $j = 1$ denote the number of rolling time periods (39 rolling 6-month aggregated time periods with monthly estimates). $i = 1$ denotes the incidence area (5259 census block group centroids, 10376 ubiquitous group centroids), $y_{ij}$ is the number of incident syphilis cases and $n_{ij}$ is the population at time $t_j$. Time periods that did not contain incident syphilis reports were assumed to have a rate of 0 cases per 10,000. The CBG dataset is composed of these crude rates.

To construct rates for the uniform grid based method, the CBG $R_{ij}$ rates are enriched with a grid-based incidence rate field. Cases located within each of these grid-based incidence areas are identified and the underlying population within the ubiquitous area is systematically calculated and used to construct a rate assigned to each grid centroid. The uniform grid is a series of grid points constructed solely within Forsyth County.

The distance between grid points, $D$ was set using the following equation:

$D = f * \frac{1}{n}\left(\sum_{a=1}^{n} \frac{\sqrt{A_a}}{\pi}\right)$, where $n$ is the user selected number of CBGs (we used 5), $A_a$ is the area

of the $a^{th}$ CBG (i.e. 1 of 5), $\frac{1}{n}\left(\sum_{a=1}^{n} \frac{\sqrt{A_a}}{\pi}\right)$ calculates the average CBG area, and $f$ is the factor

used to increase or reduce the spatial resolution of the grid. For this study we used $f = 0.65$ to create a fine grid lattice of grid points throughout the study area.

Each uniform grid (UG) boundary is a perfect circle with a radius $r_i$, where $i$ is the spatial location of the uniform grid centroid/"point". $r_i$ is calculated using an inverse weighted distance area average of the five closest CBGs to $r_i$. It is shown as, $r_i = \sum_{a=1}^{5} d_{ia}^{-1} * \frac{\sqrt{A_a}}{\pi} / \sum_{a=1}^{5} d_{ia}^{-1}$ where $d_{ia}$ is the distance between the grid point and its $a^{th}$ CBG neighbor. This approach allows for a gradual change in the size of the uniform grid areas in relation to their local CBGs, as seen in Fig 2. The boundaries of the uniform grid overlap producing a continuous rate field that smooths the edge effect.

The population of the uniform incidence area is a summation of the proportion of populations from each census block groups it overlaps. Calculated as: $n_{ij} = \sum_{a=1}^{n} \frac{\|CBG_a \cap UG_i\|}{\|CBG_a\|} n_{aj}$. The percentage of every census block group within the uniform grid incidence area is calculated, where $\|CBG_a \cap UG_i\|$ is the area of the intersection between $CBG_a$ and uniform grid boundary $i$ $(UG_i)$, and $\frac{\|CBG_a \cap UG_i\|}{\|CBG_a\|}$ is the proportion of the area of $CBG_a$ in $UG_i$. This value is then multiplied by $n_{aj}$, the CBG population, $n$ in $CBG_a$ at time $t_j$. These population values were then summed to calculate the total population for the uniform grid incidence area, $n_{ij}$. Although this approach assumes a uniform population distribution within the CBG, the regularity of the grid centroid placement pulls populations from multiple centroids, creating an enriched, modeled population at each point smoothing some of the issues with the population homogeneity assumption.

To continue to increase the spatial accuracy of the dataset, we reverse calculated the donut geomask for each case. The number of cases within each uniform incidence area was calculated as a function of the probability a geomasked case is within the uniform grid. The following is known about the geomasked cases: 1) each case is geomasked within the CBG they are located in and 2) there is a maximum and minimum distance a case can be moved from its original location [28; 44–45] (these values can only be accessed on secure computers onsite at the NC-DHHS by selected researchers). From this we can infer the probability $w_{li}$ of the $l^{th}$ geomasked case at time $t_j$ in area $UG_i$ is $w_{li} \approx \frac{\|RDCBG_l \cap UG_i\|}{\|RDCBG_l\|}$. $RDCBG_l$ is geographic area of original donut area case $l$ was initially located in its CBG before geomasking. More specifically, $RDCBG_l = RD_l \cap CBG_l$, where $RD_l$ is the reverse geomasked donut centered on the geomasked location of case $l$, and $CBG_l$ is the Census Block Group area case $l$ is located in. Then, the uniform grid incidence rate over area $UG_i$ at time $t_j$ is calculated as $R_{ij} \approx \frac{\sum_{l=1}^{N_j} w_{li}}{n_{ij} T}$, where $N_j$ is the total number of cases at time $t_j$, and $\sum_{l=1}^{N_j} w_{li}$ is the sum of the probabilities that each case at time $t_j$ is located in $UG_i$.

## Spatiotemporal analysis and incidence mapping

Bayesian Maximum Entropy (BME) is a highly developed approach of contemporary geostatistics using a spatiotemporal analysis structure [61–69]. BME techniques based on kriging have been successfully applied in wide range of public health and environmental applications [19,28,44,61–69]. These methods generate a continuous surface of exposure (such as syphilis incidence) and can incorporate soft data modeled by a distribution (population/ small number problem) as done in the Uniform Model BME (UMBME) method. In public health analysis UMBME methods have fundamental benefits over other methods including the ability to comprehensively minimize the small number problem.

UMBME introduces a measure of variance based on population allowing the method increase the variance (i.e. uncertainty) in areas with low populations, effectively penalizing their rates and categorizing them as more unreliable in the BME estimation model. This results

in both increased visual map interpretation and more accurate mathematical predictions [28,44]. Though kriging based interpolation methods such as BME are highly accurate [19,28,44,61–69], one downside is the creation of "kriging islands" [19]. This is seen at actual data point locations within the map (i.e. CBG centroids) as higher or lower rates than the surrounding areas [19].

UMBME rates were calculated for both the CBG and uniform grid data sets to allow for model comparison. BME and UMBME methods are coupled space-time calculations that provide estimates for each location at each time period using both space and time data to populate the model. The inputs for the BME spatiotemporal model include: the data points (CBG and UG) containing rate values, the mean trend for the data (must be removed to ensure model assumption of field homogeneity), a covariance model, and the UBME rates constructed from the localized population values. The output for the BME analysis returns a space-time rate field of the calculated point estimates and their associated estimation error. These point estimates can then create time series maps of the Forsyth County syphilis outbreak [28,54,62–64].

The BME numerical processing was performed using MATLAB 7.8 [70] and the BMElib version 2.0c [71]. The code created and used in the analysis can be found at: https://mserre.sph.unc.edu/BMElab_web/mappingStudies/Syp_ForsythNC/. The BMElib code is freely available at: https://mserre.sph.unc.edu/BMElib_web/. For those who do not have programming knowledge there is also a graphical user interface version of BME (BMEGUI) that guides a user through the BME process, download it at: https://mserre.sph.unc.edu/BMElab_web/ [19].

## Cross-Validation of BME Methods

A cross-validation of the CBG and uniform grid methods was conducted to identify the most accurate method of modeling syphilis incidence rates. For an estimation method of interest, cross-validation comprises removing each observed value incrementally and allowing the model estimate the value of that point using the remaining data. An estimation error value is also calculated for each point and represents the difference between the actual data value and its estimation. To calculate the mean square error (MSE), the cross-validation errors for each point in each method are averaged. The MSE is always positive and a MSE of zero demonstrates the estimator perfectly predicts an observation. The MSE is effective for comparing the prediction ability of methods, where the method with the smallest MSE is considered the best predictor for the data set. To compare the MSE between the methods, only the CBG data points contained within the uniform grid and CBG datasets are evaluated (n = 28,290 points).

The MSE formulas are:

$$MSE_{CBG} = \frac{1}{n} \sum \left( \hat{X}_{ij}^{(CBG_i)} - \frac{y_{ij}}{n_{ij}T} \right)^2$$

$$MSE_{UG} = \frac{1}{n} \sum \left( \hat{X}_{ij}^{(UG_i)} - \frac{y_{ij}}{n_{ij}T} \right)^2$$

where $y_{ij}$ is the number of incidence cases, $n_{ij}$ is the population in $CBG_i$ at time $t_j$, $T$ is the incidence duration (i.e. 6 months), $\hat{X}_{ij}^{(CBG_i)}$ and $\hat{X}_{ij}^{(UG_i)}$ are the incidence rates estimated using the CBG and uniform grid methods. A mean squared error rate is calculated for each point in the dataset. These are summed and $n$ is the number of observed rates across the CBGs and observation times.

However because of the small number problem, latent rate theory is introduced and used to more effectively compare model MSEs [28,54]. Latent rate theory states as a location's population approaches infinity, the observed rate reaches the latent also referred to as the true rate.

This can also be interpreted as the rate if the small number problem was not present [28,53–54]. The latent incidence rate of disease in a given region can be defined as: $X_{ij} = \lim_{n_{ij} \to \infty} \left( \frac{y_{ij}}{n_{ij} T} \right)$ where $X_{ij}$ is the latent disease rate, $y_{ij}$ are the incidence cases and $n_{ij}$ is the population. In practice, you can very simply estimate the latent rate by stratifying the MSE result for each point in a model by its population. To calculate each point's MSE values and their corresponding populations are categorized into population percentiles and an average MSE is calculated for each population percentile (i.e. 10, 20...) [28]. Finally, the difference between the average MSE for the two models at the population percentiles are calculated, this quantifier is referred as the percent change in the mean square error (PCMSE) [28,53–54].

With PCMSE, the method that more accurately predicts the latent rate will result in both a negative PCMSE and its negative PCMSE will remain stable or become increasingly negative as population percentile increases [28; 53–54]. The PCMSE is calculated by:

$$PCMSE_{UG} = 100 * \frac{MSE_{UG} - MSE_{CBG}}{MSE_{CBG}}$$

## 3. Results

Our cross-validation demonstrated the uniform grid model performed noticeably better in predicting the latent rate than the CBG model minimizing the mean squared error of the predictions. Fig 3 graphs the PCMSE between the two methods showing a clear decrease in the PCMSEUG as the population percentile increases. This demonstrates the uniform grid method is a better predictor of the latent rate, predicting syphilis rates 5% to 26% more accurate than the traditional, geopolitical based, CBG method.

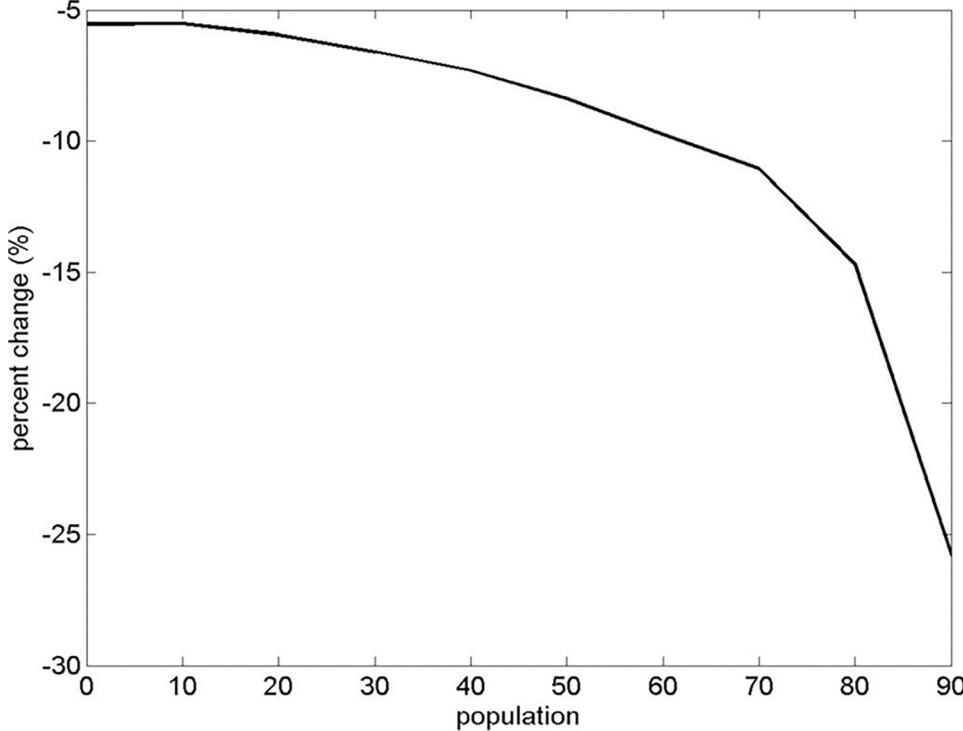

**Fig 3. The percent change in MSE from CBG to uniform grid as a function of the population percentile.**

In addition to the mathematical increases in prediction accuracy differences in the methods are visible in the maps of the two approaches (Fig 4). The uniform grid method shows increased connectivity between hotspots compared to the CBG method (Fig 4, Boxes 1, 2, 3 and 5). The uniform grid method also led to identification of an additional hotspot (Fig 4 and Box 5). In the CBG map, Box 5, a small hotspot is misplaced outside of the Winston-Salem city limits, likely produced as a kriging island. In contrast, in Box 5 the uniform grid method reduces the kriging island effect showing instead a pronounced hotspot within the Winston-Salem city limits. New hotspots also appear in the uniform grid map and are shown in all the Boxes in Fig 4, particularly in areas between spatial aggregations. Furthermore the interconnectedness of hotspots is changed in both maps demonstrating the edge effect in the CBG map.

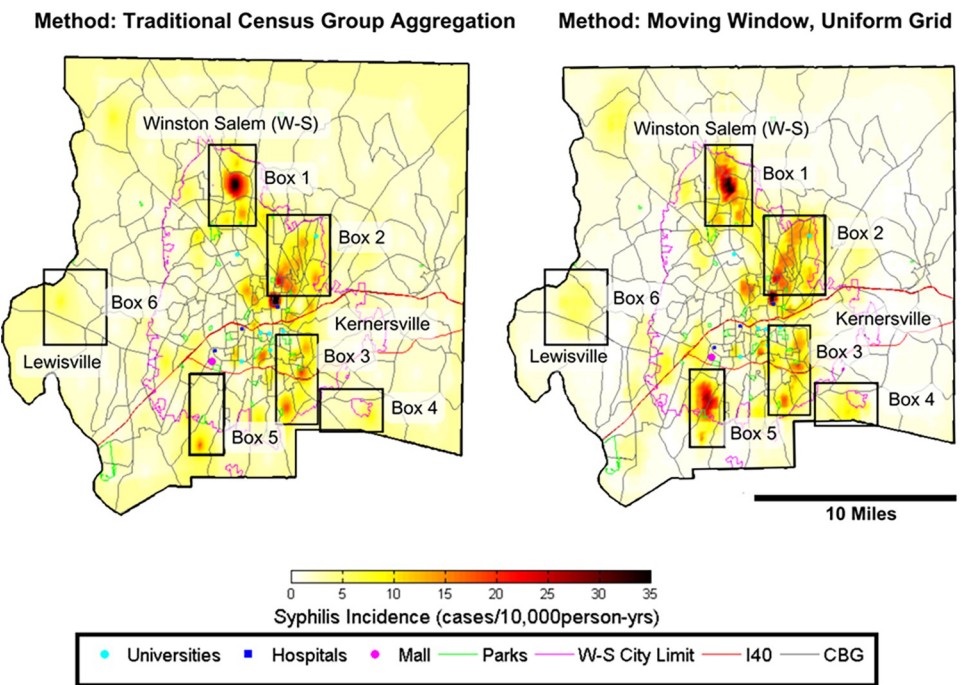

**Fig 4. BME maps of the CBG & uniform grid methods in February-July, 2009 (the peak of the outbreak).** The boxes highlight the differences in the maps showing increased connectivity, hotspot locations moved, and a reduced background rate in the Moving Window, Uniform Grid Map. Census Block Group Boundaries are publically available by the US Census at: census.gov/geographies/mapping-files/time-seriesgeo/carto-boundaryfile.2000.html#list-tab-U2GTBOCTYRAP13D9AH.

Additionally, the background rate in the CBG map is approximately 5–9 cases/10,000person-years, indicated by the overall darker yellow color for the whole county and likely the result of the kriging islands effect. The uniform grid method has reduced the kriging island effect allowing a truer background rate of 0–2 cases to be shown. This permits more minor hotspots to appear in the rural areas of the map highlighting rural areas of concern shown in Boxes 4 and 6.

Outbreak information can also be gained from the covariance model produced in the BME analysis. The covariance model describes the geographic and temporal features of the dataset. The spatial component of the covariance model and it's the experimental covariances values (top panel of Fig 5), indicate syphilis outbreak hotspots are generally concentrated to small areas (less than 10% of the study area). Temporally, the covariance shows hotspots persisting for more than 6 months up to the duration of the Forsyth outbreak. This shows that resources could potentially be targeted to small areas that persist for relatively long time periods.

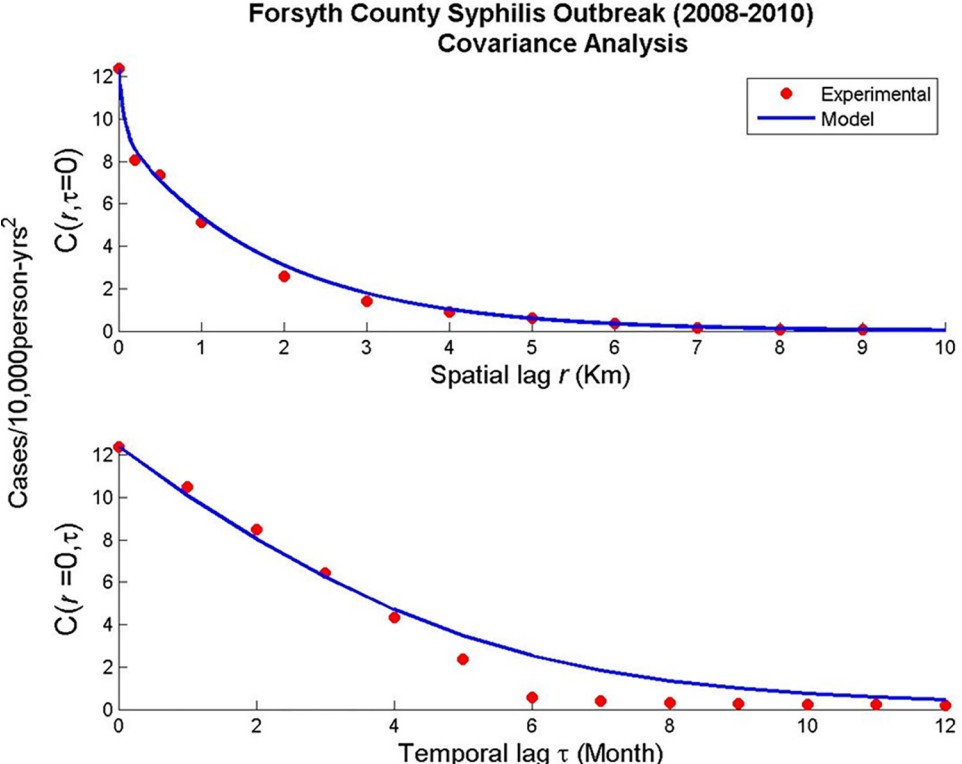

**Fig 5. Spatial and Temporal Covariance Models.**

BME maps show the pattern of outbreak development (Fig 6; movie also available at https://mserre.sph.unc.edu/BMElab_web/mappingStudies/Syp_ForsythNC/RevisedUMBMEMovieForsythGrid.gif). Across Forsyth County, areas with an incidence of approximately 15 cases/10,000person-years are identifiable throughout the southeastern region of Winston-Salem and lower rates of 2-5cases/10,000person-years are found within Lewisville and Kernersville.

**Fig 6. Uniform grid method time series of the Forsyth County Outbreak (2008–2010).** Census Block Group Boundaries are publically available by the US Census at: census.gov/geographies/mapping-files/time-seriesgeo/carto-boundaryfile.2000.html#list-tab-U2GTBOCTYRAP13D9AH.

Rapid incidence increases to ≥20 cases/100,000 person-years within small geographical boundaries cumulatively lead to an outbreak that began to spread. In the aggregated time period of June-November, 2008 small hotspots begin to appear and increase; some portions exhibit rates at or above 35 cases/10,000person-years. In September, 2008-February, 2009 the hotspots continue to spread throughout Winston-Salem, increasing in intensity and connection in the north central and north eastern portions of the city. As the outbreak progresses, new hotspots appear in the eastern region of Winston-Salem and begin to connect (Fig 6 and video). The peak of the outbreak is February-July, 2009. As the outbreak wanes, the hotspot in the north-central region of the city is reduced, while a hotspot in northeast increases. This hotspot shows increased connection in April-September, 2009 specifically in central-east Winston-Salem near I-40 and in the northeast regions. These hotspots continue to grow and connect until May-October, 2009 then wane and disconnect.

In November, 2009-April 2010, the rates begin to return to endemic levels. In 2010–2011 the outbreak continues to decrease with increasing disconnection of the hotspots. By February-July, 2010 the Winston-Salem incidence has returned to endemic levels of 3–7 cases per 100,000 and the outbreak has subsided. Endemic is when the incidence is low and stable, and was defined in a study by Hook in the Lancet in 1998 as an area with 4 or more syphilis cases per 100,000 [72]. Furthermore, the Kernersville and Lewisville syphilis rate returns to 1–5 cases 10,000person-years. Fig 6 and the movie display the progression of the syphilis epidemic across Forsyth County between 2008–2010 with its three largest towns labeled (Winston-Salem, Lewisville, and Kernersville).

## 4. Discussion

The methods presented in this article advance outbreak analysis by addressing common issues in sexually transmitted infection (STI) mapping creating more accurate and more easily interpretable maps. Conducting a spatiotemporal case-study review of the syphilis outbreak in Forsyth County, North Carolina in 2008–2011 allowed insight into how the outbreak emerged from endemic areas, developed and then subsided. We show combining a moving window

and a UMBME analysis with geomasked data produced more realistic geographical patterns for more accurate and targeted insights into a syphilis outbreak. The improved visualization in the maps was accompanied by less apparent error, with the uniform grid model performing 5–26% better in predicting the true rate compared to the traditional, geopolitical CBG based model.

Our approach is particularly novel because we introduce the ability to remove what is commonly referred to as "kriging islands", where rates are higher and lower at centroids [19] providing greater sensitivity for hotspot detection. This enhanced visualization is especially important when hotspots are located on or across the boundary of two or more aggregation areas which traditionally can result in the complete loss of a hotspot. Furthermore, removing kriging islands reduces the background rate which unmasks rural hotspot areas of concern. Thus, removing kriging islands resulted in an increased ability to visualize spatial corridors of incidence and a more accurate representation of interconnection and spread. Throughout the outbreak, our methods placed incidence hotspots within the Winston-Salem city limits, rather than at administrative centroids which are often outside the city limits or in underdeveloped, sparsely populated areas.

This Forsyth outbreak in 2008–2011 is similar to the 2001 Robeson/Columbus, North Carolina syphilis outbreak [19], and began in areas with endemic syphilis. As the endemic hotspots intensified they also spread and leap-frogged to connect into non-endemic areas. In both outbreaks, the hotspots were highly localized and generally persisted for one year [19]. This endemic leap-frogging is also similar to what was seen in the Baltimore, Maryland syphilis outbreak in 2002 [67]. The Forsyth outbreak slightly contrasted to the Robeson/Columbus outbreak in 2000–2002, because in Forsyth the cases were primarily contained to the urban, endemic areas of Winston Salem; whereas in the Robeson/Columbus outbreak the cases leaped from the cities to the rural areas [19]. Additionally, we found many of the hotspots in the Forsyth outbreak were located near or in between popular meet up areas, such as parks, highways, and popular malls in Winston Salem. This is similar to past outbreaks which have moved along the Interstate Highway 95 [13].

Other work has also shown cases with partners from many locations can act as spatial spread/network bridges, geographically distributing the disease [12]; this bridging effect appeared to be occurring in this outbreak as well. In a geographic STI partner study in Toronto in 2016, 78% of their urban-outbreak sexual network participants remained in the outbreak/hotspot areas to find new partners and rarely travelled to the suburban or rural areas for partner selection. However, 81% of those living in the rural areas and suburbs travelled into the urban outbreak areas to meet new partners [73]. This may have been occurring in this outbreak as well, as the bulk of the cases in the 2008–2011 Forsyth outbreak were isolated to the urban areas, however there are some isolated cases in the more suburban and rural areas near Lewisville and outside of Winston-Salem. Removing the kriging islands allowed the identification of these rural hotspot locations, which otherwise would have been masked within a homogenous background rate.

Some of the dynamics of syphilis transmission have changed since 2008. First, in the US overall the rate of primary and secondary syphilis for men who have sex with men (MSM) has decreased from 72% in 2010 [74] to 45% in MSM with an additional 20% in men who have sex with unknown partners (MSU) in 2021 [2]. However, some of the 2021 MSU data likely could be categorized within MSM [75–76]. Conversely, in North Carolina in 2022 the early syphilis rate for gay, bisexual and other men who have sex with men was estimated to be 1,054 per 100,000 whereas the syphilis rates of: 1) men who only have sex with women and 2) women were both 17 per 100,000 [29]. As 85% of syphilis cases in 2008 were in men [16], in North Carolina the primary high risk population may not have significantly changed. Additionally many of

the rates seen in the 2008–2011 Forsyth outbreak [16] are similar to 2021 syphilis rates in the North Carolina, where the county rates in the 20 most affected counties ranged from 54.8–23.5 syphilis cases per 100,000 [14]. The gender change in syphilis transmission in the US in general from 2011 to 2021 could affect the application of the results of geographical transmission shown in this study to outbreaks outside of North Carolina.

Additionally methods for partnership selection have changed substantially since 2011. In 2008-

2011 people often "cruised", i.e. meeting partners at physical environments (e.g., bath-houses, parks, parks, pools, malls). With the wide-spread adoption of internet based phones, most people now rely on apps and online dating websites for partner selection changing the community level risk space for syphilis [73,77–80]. However studies have also shown partici-pants who have a more challenging time with dating apps still frequent common activity spaces to meet partners [73,80].

Yet even in the dating app era, localized geographic proximity is still a key factor partner selection for both men and women [73,80–81]. In practice, distance travelled to meet sexual partners may not have drastically changed over time. Interviews on partner selection have shown participants preferentially selected partners for whom they had to travel the shortest distance possible to connect [73,80]. Research continues to support that partnership selection is a localized event whether this selection occurs online or offline [73,80–83]. Therefore many of the assumptions around geographical partner selection dynamics shown in this work may remain in 2024.

There are a few other limitations of our study. First, we focused on a single county and sin-gle outbreak. Primary and secondary syphilis cases are heavily geographically concentrated [19–20,27,84], more than other STIs such as chlamydial infection and gonorrhea [84–85]. Although our findings mirror other syphilis outbreaks [13,19–20,27,67], use of a different geo-graphical area, such as an entire state or country, may have led to different conclusions [19–20,67,27]. Advances in BME techniques, big data methods and optimization have recently to allow BME analysis to be performed on large domains, such as Kenya [86] with 2 billion data points and the continental United States [87]. In the past, BME studies required a relatively small spatiotemporal dataset for computation. Second, the combination of the UMBME method with a moving window and regular grid is ideally suited for relatively rare conditions, such as syphilis. The potential benefits of this approach for more common infec-tions, such as chlamydial infection and gonorrhea, are uncertain [84–85].

Our approach can benefit health authorities by providing greater resolution of ongoing out-breaks, allowing for targeted intervention and resource allocation [12,19,27,44,65,73]. Even if the spatial dynamics of STI transmission have changed it is still important to have highly accu-rate and easily interpretable visual representations of STI data. Understanding the spatial pro-gression of outbreaks can help to contextualize current and past outbreaks and help describe the spread within and outside sexual core transmitters and their sexual network [19,73]. These insights can provide public health officials a clearer view of the sexual network location and their potential activity spaces (where the sexual behaviors are actually occurring) [12,19–20,27,73] to isolate areas of need and inform intervention strategy selection and development [73].

## Author Contributions

**Conceptualization:** Lani Fox, William C. Miller, Dionne Gesink, Marc Serre.

**Data curation:** Lani Fox, William C. Miller.

**Formal analysis:** Lani Fox, Marc Serre.

**Funding acquisition:** William C. Miller, Dionne Gesink, Irene Doherty, Marc Serre.

**Investigation:** Lani Fox, Marc Serre.

**Methodology:** Lani Fox, William C. Miller, Marc Serre.

**Project administration:** William C. Miller.

**Resources:** William C. Miller, Marc Serre.

**Supervision:** William C. Miller, Marc Serre.

**Validation:** Lani Fox.

**Visualization:** Lani Fox.

**Writing – original draft:** Lani Fox, William C. Miller, Marc Serre.

**Writing – review & editing:** Lani Fox, William C. Miller, Dionne Gesink, Irene Doherty, Marc Serre.

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
