## [Decision Letter · Decision Letter 0]

12 Apr 2024

Dear Mrs Fox,

Thank you very much for submitting your manuscript "Reducing ecological bias and improving insights in sexually transmitted infection outbreak mapping: Syphilis in Forsyth County, a case study" for consideration at PLOS Computational Biology.

As with all papers reviewed by the journal, your manuscript was reviewed by members of the editorial board and by several independent reviewers. In light of the reviews (below this email), we would like to invite the resubmission of a significantly-revised version that takes into account the reviewers' comments.

I agree with the reviewers that this is a potentially interesting contribution related to a persistent (likely growing), but understudied, sexually transmitted infection in the United States. However, I share the reviewer concerns about whether data from almost 15 years ago is still relevant for understanding current trends. My advice to the authors is that a strong focus of the revision needs to be on defending the use of this time period. If more recent data are not available for the study area, perhaps adding some plots about state- and national-trends to help contextualize where things were in 2008-11 vs today? I want to stress that successfully addressing the reviewer comments really hinges on this particular issue.In addition, I also agree that the authors should take care during the revision to clarify the specifics of what was done, assumed in the model, and found via analysis of the model. Particular focus should be placed on linking terminology/approaches from geospatial ecology to infectious disease epidemiology. As the authors understand, conceptually the approaches are quite similar, but the jargon will be a barrier to an interdisciplinary read/understanding of this work. Finally, please ensure you share the code the analysis in compliance with our code-sharing policy.

We cannot make any decision about publication until we have seen the revised manuscript and your response to the reviewers' comments. Your revised manuscript is also likely to be sent to reviewers for further evaluation.

Sincerely,

Samuel V. Scarpino

Academic Editor

PLOS Computational Biology

Virginia Pitzer

Section Editor

PLOS Computational Biology

I agree with the reviewers that this is a potentially interesting contribution related to a persistent (likely growing), but understudied, sexually transmitted infection in the United States. However, I share the reviewer concerns about whether data from almost 15 years ago is still relevant for understanding current trends. My advice to the authors is that a strong focus of the revision needs to be on defending the use of this time period. If more recent data are not available for the study area, perhaps adding some plots about state- and national-trends to help contextualize where things were in 2008-11 vs today? I want to stress that successfully addressing the reviewer comments really hinges on this particular issue.In addition, I also agree that the authors should take care during the revision to clarify the specifics of what was done, assumed in the model, and found via analysis of the model. Particular focus should be placed on linking terminology/approaches from geospatial ecology to infectious disease epidemiology. As the authors understand, conceptually the approaches are quite similar, but the jargon will be a barrier to an interdisciplinary read/understanding of this work.

Reviewer's Responses to Questions

**Comments to the Authors:**

Reviewer #1: The authors present a spatio-temporal analysis of reported early syphilis in Forsyth County, NC from 2008-2011. Early syphilis cases were geocoded and several methodologic approaches were applied to the data to describe the spatio-temporal distribution, in an effort to reduce ecologic bias common in spatial analyses of administrative areas (counties, states, block groups). The analysis is thought provoking, however the submission lacks some clarity in its stated goals. It was not clear how the analysis demonstrated that the moving window and UMBME was the better approach. Additionally, the data used are more than a decade old which may have limited the utility of the analysis.

General comments:

-Age of data used: the data included in this analysis were from 2008-2011. This is more than 13 years old. I also worry on including 4 years is insufficient to address temporal complications in the data. While comparing methods on older data is not inherently problematic, I worry that application of this approach on more recent data may have some issues. Syphilis rates have increased broadly since 2011 and the epidemic is much less MSM focused. Additionally, the wide-spread adoption of online and cell-phone based apps for meeting partners has changed the community level risk space for syphilis. As a result, some of the assumptions around residential geography may not be the same as in 2008-2011. For example, fewer people may meet partners at physical community sites (e.g., bars, parks, schools) and rely more on app based approaches – this would alter the way geography may play into syphilis risk. Additionally, since the data was collected, the underlying population may have changed geographically in important ways. For example, has the distribution of MSM changed in the geographic area? This would alter the findings and modify the spatial risk, which may result in different findings. While I understand more contemporary data may not be available, I encourage the authors to explain more about: (1) why the older data were used, (2) the reasons why more recent data may not be available, and (3) what are the potential implications in translating their findings data collected more recently.

-Similar analyses to others presented by this group: The author team is well suited for analyses like this and are well published in this area. I do however wonder what this analysis adds to the literature beyond their prior work. A similar analysis has been recently published in PLOS Global Health and other papers have been published over the past decade. I encourage the authors to address how this analysis is different and novel and what it adds to the literature beyond their prior stellar work.

-Improved clarity in analytic goals and results: From my reading, the goal of this analysis was to examine different spatial methods to reduce bias from ecological fallacy and edge effects. This was not super clear and took some digging around to figure this out. The way the results are organized, this goal is cursorily addressed and not very clearly. I recommend the authors review and make clear the goals and how the results meet these goals.

Specific comments:

-Background – some important concepts are not very well described. For example, lines 83-5 these should be explained more clearly to an audience who may not have a depth of knowledge in spatial work.

-Lines 94-5 what is meant by “connectivity” and “geographical extent”?

-Line 145- why was the CDC IRB review conducted?

-Lines 150-7- I did not follow the three geographic indicators and how they were used to plot one location?

-Lines 158-66 – were comparisons made between early syphilis cases that could and could not be geocoded? Any important differences?

Line 174- Is ref 44 correct here?

Lines 177-92- I found this section quite unclear. Figure 1 is not great resolution and I did not understand what was solid turquoise circle.

Lines 200-252- I also did not follow how denominator data was obtained for the circle approach. CBG have known populations to use as denominators, how were these used to estimate the incidence rates for these area? Doesn’t the approach used basically imply the same issues as edge effects and ecologic fallacy? It assumes a uniform population within a CBG which is not likely, so how does this approach help address the issues brought up in the intro?

Lines 329- The text reads Aug 2008-Mar 2010, but other places (like the abstract) say data were used from 2008-2011. This needs to be clarified and consistent throughout.

Line380-1: The statement about endemic syphilis being <=15 cases/100,000 needs a reference. I have never heard this before.

Figure 1: the labels in the text don’t match the figure, what is ubiquitous and arbitrary mean here?

Line 424-426: I don’t think the results presented show this.

Lines 438-440- data presented don’t show this.

Reviewer #2: Thank you for the opportunity to review this manuscript. The authors present their novel approach to assess spatiotemporal infectious syphilis data. With national P&S syphilis rates reaching levels not seen in decades, this is an important research area and a timely study. Overall, the authors thoroughly describe an interesting approach to low-rate disease surveillance that could be of use to local public health entities. The statistical approaches utilized are rigorously documented, validated and appear appropriate for the problem described. I have several suggestions that could help improve this manuscript.

Overall:

• The dates of cases included in the study are unclear to me as different date ranges are described at different points in the paper. While this is appropriate when describing the rolling windows in the results and the date range of the specific outbreak, in other locations in the manuscript it raises confusion: eg. In the methods, are all syphilis diagnoses between 1999-2011 included or is it 2008-2011 as is stated elsewhere. Suggest clarifying the differing date ranges listed throughout the manuscript

Methods:

• Suggest either limiting to P&S syphilis cases or describing potential limitations of including early latent syphilis cases. While this is unlikely to be an issue for this study as it focuses on a single county, the formal case definition of early latent syphilis was known to cause confusion when staging the case, leading to potential misstaging and issues with generalizing between locales, so generally, reporting on syphilis surveillance is restricted to P&S cases. Additionally, this stage has since been revised to early non-primary non-secondary (ENPNS) syphilis with an updated case definition

• Additional descriptives could help with the interpretation of later results. For example, a table detailing # of cases by stage and/or year and/or # of excluded cases by stage/year

• Back-dating to estimate syphilis transmission dates: I’m unfamiliar with these specific date ranges and the provided citation- (44. Choi KM, Serre ML, Christakos G. Efficient mapping of California mortality fields at different spatial scales. Journal of Exposure Science & Environmental Epidemiology. 2003 Mar;13(2):120-33) doesn’t seem to mention syphilis

Results:

• Provide a citation for rate definition of endemic syphilis (lines 379-381)

Discussion:

• I would be interested in some clarification of the implications of the finding that many hotspots seem to be located near popular meeting locations (lines 438-440 and 455-457)

• (Not a suggestion) I appreciate the description of limitations of this approach

Reviewer #3: I would like to congratulate the authors for the paper, it is essential that we can advance in outbreak analysis methods, improving the decision-making process in risk mitigation.

Overall, in my opinion, the article is very well structured and well-founded.

However, I list points in the methodology that need improvement:

- Present references for choosing the quantities described between lines 218 and 223. In particular: when D=0.5 is chosen, this is the value of the equation presented in line 219, but the average CBG radius is not defined?

- The word radii in line 226 was not clear to me.

- Present references for how calculate the number of cases within each uniform incidence area.

- The quantity Wli, defined in line 245, needs to be better explained. How to calculate Reverse Donut area? Wouldn't it be more appropriate to count the number of cases in each UG, since the occurrences are already geomasked?

**Have the authors made all data and (if applicable) computational code underlying the findings in their manuscript fully available?**

Reviewer #1: Yes

Reviewer #2: **No: **The authors note that the data utilized are PHI and would require a formal data request to obtain

Reviewer #3: **No: **The code must be available so that the methodology can be replicated in other circumstances.

PLOS authors have the option to publish the peer review history of their article (what does this mean?). If published, this will include your full peer review and any attached files.

Reviewer #1: No

Reviewer #2: No

Reviewer #3: No
---

## [Decision Letter · Decision Letter 1]

5 Sep 2024

Dear Mrs Fox,

We are pleased to inform you that your manuscript 'Enhancing insights in sexually transmitted infection outbreak mapping: Syphilis in Forsyth County, North Carolina case study' has been provisionally accepted for publication in PLOS Computational Biology.

Best regards,

Samuel V. Scarpino

Academic Editor

PLOS Computational Biology

Virginia Pitzer

Section Editor

PLOS Computational Biology

Reviewer's Responses to Questions

**Comments to the Authors:**

Reviewer #1: All my concerns have been addressed

Reviewer #2: Thank you for your responses. This reviewer is satisfied with the edits and additions provided by the authors.

Reviewer #3: With the changes implemented in the review, the text addresses the points raised by the reviewers.

**Have the authors made all data and (if applicable) computational code underlying the findings in their manuscript fully available?**

Reviewer #1: Yes

Reviewer #2: **No: **These data require a formal data request as they contain PHI

Reviewer #3: Yes

PLOS authors have the option to publish the peer review history of their article (what does this mean?). If published, this will include your full peer review and any attached files.

Reviewer #1: No

Reviewer #2: No

Reviewer #3: No

---

## [Editor Report · Acceptance letter]

25 Oct 2024

PCOMPBIOL-D-24-00363R1 

Enhancing insights in sexually transmitted infection mapping: Syphilis in Forsyth County, North Carolina, a case study

Dear Dr Fox,

I am pleased to inform you that your manuscript has been formally accepted for publication in PLOS Computational Biology. Your manuscript is now with our production department and you will be notified of the publication date in due course.

With kind regards,

Anita Estes
